# Status and Developing Strategies for Neutralizing Monoclonal Antibody Therapy in the Omicron Era of COVID-19

**DOI:** 10.3390/v15061297

**Published:** 2023-05-31

**Authors:** Zuning Ren, Chenguang Shen, Jie Peng

**Affiliations:** 1State Key Laboratory of Organ Failure Research, Guangdong Provincial Key Laboratory of Viral Hepatitis Research, Department of Infectious Diseases, Nanfang Hospital, Southern Medical University, Guangzhou 510515, China; renzuning@163.com; 2BSL-3 Laboratory (Guangdong), Guangdong Provincial Key Laboratory of Tropical Disease Research, School of Public Health, Department of Laboratory Medicine, Zhujiang Hospital, Southern Medical University, Guangzhou 510515, China

**Keywords:** SARS-CoV-2, COVID-19, monoclonal antibody, Omicron variant

## Abstract

The monoclonal antibody (mAb)-based treatment is a highly valued therapy against COVID-19, especially for individuals who may not have strong immune responses to the vaccine. However, with the arrival of the Omicron variant and its evolving subvariants, along with the occurrence of remarkable resistance of these SARS-CoV-2 variants to the neutralizing antibodies, mAbs are facing tough challenges. Future strategies for developing mAbs with improved resistance to viral evasion will involve optimizing the targeting epitopes on SARS-CoV-2, enhancing the affinity and potency of mAbs, exploring the use of non-neutralizing antibodies that bind to conserved epitopes on the S protein, as well as optimizing immunization regimens. These approaches can improve the viability of mAb therapy in the fight against the evolving threat of the coronavirus.

## 1. Introduction

Coronavirus Disease 2019 (COVID-19), caused by Severe Acute Respiratory Syndrome Coronavirus 2 (SARS-CoV-2), which belongs to the subgenus sarbecovirus of the beta genus of coronaviruses, was declared an international Public Health Emergency of International Concern (PHEIC) by the World Health Organization (WHO) in January 2020, with more than 757 million people infected and 6.8 million people died [1].

Although the number of severe COVID-19 cases has decreased in 2023, immunocompromised patients, such as organ transplant recipients, individuals with cancer, advanced or untreated HIV, and autoimmune disorders, as well as those from high-risk groups (age over 65, obesity or overweight, pregnancy, chronic kidney disease, diabetes mellitus, cardiovascular disease, hypertension, chronic lung disease, etc.), still require antibody treatment and pre-or post-exposure prophylaxis [2].

Monoclonal antibody (mAb)-based treatment is a highly valued therapy, especially for individuals who may not have strong immune responses to the vaccine, such as elderly or immunocompromised patients [3]. The mAbs are recombinant synthesized molecules that may prevent the interaction between the viral envelope and cell receptors or inhibit the release of the viral genome, thus, preventing viral entry and replication and subsequently protecting host cells from infection [4]. In addition to neutralization, some mAbs can induce clearance of virus-infected cells through antibody-dependent cellular cytotoxicity (ADCC), antibody-dependent cellular pathogenicity (ADCP), and complement-dependent cytotoxicity (CDC), thereby eliminating the virus [5,6,7].

SARS-CoV-2 possesses four distinct structural proteins, including the envelope (E) protein, membrane (M) protein, spike (S) protein, and the nucleocapsid (N) protein. The replication, pathogenicity, and dissemination [8] of the virus are modulated by the E protein, while the M protein is accountable for the assembly of viral particles [9]. The S protein facilitates the penetration of the virus into the host cell [10], while the N protein assembles the viral genomic RNA into a ribonucleoprotein complex and regulates viral replication [11]. 

The S protein of SARS-CoV-2 is a glycoprotein on the coronavirus surface, consisting of an S1 subunit and an S2 subunit that, respectively, mediate attachment and membrane fusion. The receptor-binding domain (RBD) and the N-terminal domain (NTD) are two independent domains on the S1 subunit [12]. RBD mediates viral invasion of host cells by strongly binding to angiotensin-converting enzyme 2 (ACE2) on the host cell surface [10,13,14], making it a key target of mAbs treatment of coronavirus infection [15,16,17,18]. Prior studies have found that ACE2 competitive antibodies occupy most of the RBD-specific antibodies induced by natural infection or vaccination [19], and this strategy is also applied in mAbs development. Residues 424 to 494 of the RBD, which have been designated the receptor-binding motif (RBM), play a critical role in inducing neutralizing antibodies during the process of ACE2 interaction [10,20]. The N-terminal domain (NTD) in the S protein, which has been believed to be related to controlling the S protein stability, membrane fusion ability, and RBD exposure [21,22], is also a considerable target for coronavirus. However, its variability is relatively higher compared to other parts of the virus, resulting in a limited application potential for the antibodies it induces. Therefore, mAbs targeting the S protein, especially the RBD, remain one of the most promising approaches to developing effective mAbs against SARS-CoV-2.

Several neutralizing mAbs have been authorized for emergency use by the European Medicine Agency (EMA) and the American Food and Drug Administration (FDA) between 2021 and 2022, such as sotrovimab [23], bebtelovimab [24], and the cocktails REGEN-COV2 (casirimab/imdevimab) [25], bamlanivimab/etesevimb [26], and Evusheld (tixagevimab/cilgavimab) [27]. These mAbs had shown high effectiveness in reducing hospitalization rates and improving clinical outcomes, however, they had been announced to be excluded from Emergency-Use-Administration (EUA) as they cannot provide protection from current SARS-CoV-2 circulating variant infections [28]. 

While mAbs have been a promising treatment against COVID-19, the emergence of new variants, such as the Omicron variant and its subvariants, have led to concerns regarding their efficacy. Thus, optimizing strategies for mAbs development and construction has become a pressing issue. This review aims to introduce the challenges faced by therapeutic neutralizing mAbs in the era of Omicron variants and explore strategies for optimizing their development and construction.

## 2. Main Text

### 2.1. Most Existing Therapeutic mAbs Targeting the RBD Have Lost the Ability to Neutralize the Current Omicron Variants

In November 2021, the Omicron (B.1.1.529) variant was initially discovered and promptly classified as a variant of concern (VOC) by the WHO [29]. Since then, it has rapidly increased in frequency worldwide, and it currently accounts for over 98% of the viral sequences shared on the Global Initiative of Sharing All Influenza Data (GISAID) [30]. While descendent lineages continue to evolve, only the subvariant XBB.1.5, a descendent subvariant of XBB, has currently reached high endemic levels globally. Since 23 January 2023, XBB.1.5 has surpassed BQ.1.1 as the predominant circulating variant in the USA and currently accounts for 85.0% of infections (Figure 1). 

Like other RNA viruses, as SARS-CoV-2 evolves, it accumulates mutations that can result in the evasion of neutralizing antibodies and alterations in its binding to host receptors [32]. The Omicron variant’s genetic and antigenic dissimilarity from the original SARS-CoV-2 is significant, leading to remarkable evasion of antibodies [33]. The RBD in the S protein is particularly important for neutralizing antibody responses, with more than 90% of these antibodies targeting this region [19]. Mutations on the RBD have been demonstrated to lead to evasion from neutralizing antibodies [34,35]. The RBD changes affected binding surfaces for the ACE2 receptor as well as recognition sites for potently neutralizing antibodies. In the meantime, the modifications in local conformation and hydrophobic micro-environments induce changes in epitopes, rendering Omicron resistant to the majority of RBD and NTD antibodies [33]. Multiple mutations are likely needed for simultaneous escape from most antibodies, but it is unclear how these mutations might interact [36]. Variations in therapeutic epitopes or binding sites (VinTEBS) on the S protein have been identified (Table 1), and the RBD (amino acids 319 to 541) sequences have been compared among currently available SARS-CoV-2 variants (Table 2). Evidently, over half of the mutations were found to affect the epitopes of RBM-binding antibodies. As variants continue to evolve, the mutations in therapeutic epitopes on the RBD accumulate gradually, which may affect the effectiveness of the neutralization of mAbs. 

A study revealed that merely 15% of the 247 human RBD mAbs could bind with the Omicron variant [37], and the vast majority of mAbs targeting the RBD had lost binding affinity and neutralizing activity against the subvariants [38,39]. However, some mAbs, such as S3H3 [40] targeting SD1, S309 (sotrovimab) [41] targeting the RBD, and C1717 [42] targeting NTD-SD2 still retain some neutralizing activity against these two subvariants relative to previous subvariants. The ineffectiveness of monoclonal antibody therapy for immunocompromised patients is a tragic reality, leaving them with very limited options when confronting SARS-CoV-2 infections [43].
viruses-15-01297-t001_Table 1Table 1Amino acid sequence of the S protein RBD. VariantsAmino Acid Sequence of S protein RBD
Region of RBM339346356368371373375376405408417440444445446452455460477478484486490493496498501505519AlphaB.1.1.7GRKLSSSTDRKNKVGLLNSTEFFQGQYYHBetaB.1.351GRKLSSSTDR**N**NKVGLLNST**K**FFQGQYYHDeltaB.1.617.2GRKLSSSTDRKNKVG**R**LNS**K**EFFQGQ**N**YHOmicronBA.1**D**RKL**L****P****F**TDR**N****K**KV**S**L**X**N**N****K****A**FF**R****X****R**Y**H****X**BA.2**D****T**KLSSSTDRKNKVGLLNSTEFFQGQ**N****Y**HBA.2.12.1**D**RKL**F****P****F****A****N****S****N****K**KVG**Q**LN**N****K****A**FF**R**G**R**Y**H**HBA.4**D**RKL**F****P****F****A****N****S****N****K**KVG**R**LN**N****K****A****V**FQG**R**Y**H**HBA.4.6**D****T**KL**F****P****F****A****N****S**K**K**KVG**R**LN**N****K****A****V**FQG**R**Y**H**HBA.5**D**RKL**F****P****F****A**DRKNKVGLLN**N****K****A****V**FQG**R**Y**H**HBA.5.2.6**D****T**KL**F****P****F****A****N****S****N****K**KVG**R**LN**N****K****A****V**FQG**R**Y**H**HBE.1.1.1**D**RKL**F****P****F****A****N****S****N****K****T**VG**R**LN**N****K****A****V**FQG**R**Y**H**HBF.11**D****T**KL**F****P****F****A****N****S****N****K**KVG**R**LN**N****K****A****V**FQG**R**Y**H**HBF.7**D****T**KL**F****P****F****A****N****S****N****K**KVG**R**LN**N****K****A****V**FQG**R**Y**H**HBQ.1**D**RKL**F****P****F****A****N****S****N****K****T**VG**R**L**K****X****K****A****V**FQG**R**Y**H**HBJ.1**H****T**K**I****F****P****F****A****N****S****N****K**K**P****S**LL**K****N****K****A****S****S**QG**R**Y**H**HBA.2.75.2**H****T**KL**F****P****F****A****N****S****N****K**KV**S**LL**K****N****K****A****S**FQG**R**Y**H**HBM.4.1GRKLSSSTDRKNKVGLLNS**K****A****S**FQG**R**Y**H**HBM.4.1.1**H****T**KL**F****P****F****A**DRKNKVGLLN**N****K****A****S**FQG**R**Y**H**HBN.1**H****T****T**L**F****P****F****A****N****S****N****K**KV**S**LL**K****N****K****A**F**S**QG**R**Y**H**HCH.1.1.1**H****T**KL**F****P****F****A****N****S****N****K****T**V**S****R**L**K****N****K****A****S**FQG**R**Y**H**HXBB.1**H****T**K**I****F****P****F****A****N****S****N****K**K**P****S**LL**K****N****K****A****S****S**QG**R**Y**H**HXBB.1.5**Y****T**K**I****F****P****F****A****N****S****N****K**K**P****S**LL**K****N****K****A****P****S**QG**R**Y**H**HXBB.1.9**H****T**K**I****F****P****F****A****N****S****N****K**K**P****S**LL**K****N****K****A****S****S**QG**R**Y**H**H**Note:** Data available from National Center for Biotechnology Information (NCBI) [44]. Bold text represents amino acid mutations.
viruses-15-01297-t002_Table 2Table 2Key mutations in therapeutic epitopes or binding sites on the S protein.SARS-CoV-2 VariantsVariations(+) in Therapeutic Epitopes or Binding Sites (VinTEBS) on S ProteinN501YK417NE484KL452RE484QT478KQ493RG446SS371L; G496SS477N;S373P;G339D;E484A;Q498R;S375F;N440K;Y505HS371F; D405N; R408SL452QR346TF486VN460Y; N334KK444TN460KR346K; F490V; V483AG339RF486SF490SK356TV445A;V445LF486LAlpha
B.1.1.7
+−−−−−−−−−−−−−−−−−−−−−−−Beta
B.1.351
+++−−−−−−−−−−−−−−−−−−−−−Delta
B.1.617.2
−−−+++−−−−−−−−−−−−−−−−−−Omicron
BA.1
++−−−+++++−−−−−−−−−−−−−−BA.2BA.2++−−-++−−++−−−−−−−−−−−−−BA.2.12.1++−−-++−−+++−−−−−−−−−−−−BA.4BA.4++−+-+−−−-++−−−−−−−−−−−−−BA.4.6++−+-+−−−++−++−−−−−−−−−−BA.5BA.5++−+-+−−−++−−+−−−−−−−−−−BA.5.2.6++−+-+−−−++−+++−−−−−−−−−BE.1.1.1++−+-+−−−++−−+−+−−−−−−−−BF.11++−+-+−−−++−++−−−−−−−−−−BF.7++−+-+−−−++−++−−−−−−−−−−BQ.1++−+-+−−−++−−+−++−−−−−−−
BJ.1
++−−−+++−++-−−−−−+−−−−−−BA.2.75BA.2.75.2++−−−++−−++−+−−−+−++−−−−BM.4.1++−−−++−−++−−−−−+−++−−−−BM.4.1.1++−−−++−−++−+−−−+−++−−−−BN.1++−−−++−−++−+−−−+−−−++−−CH.1.1.1++−+−++−−++−+−−++−++−−−−XBBXBB.1++−−−+−−−++−+−−−+−+++−+−XBB.1.5++−−−+−+−++−+−−−+−+++−++XBB.1.9++−−−+−+−++−+−−−+−+++−+−Note: Data are available from NCBI [45]. The plus sign (+) represents the presence of a mutation, while the minus sign (−) represents the absence of a mutation.

In summary, the extensive mutations on the S protein of the Omicron variant have resulted in a marked resistance to therapeutic neutralizing mAbs and vaccine-induced neutralization responses. As a consequence, all clinically authorized therapeutic mAbs targeting the Omicron variants, especially the BQ and XBB subvariants, have been rendered ineffective [28]. This highlights the urgent need to develop broadly effective mAbs that can combat the ongoing COVID-19 pandemic, as well as the importance of exploring novel strategies for developing neutralizing mAbs that can resist viral evasion.

### 2.2. Strategies for Neutralizing mAbs Development Confronting the Viral Evasion of Omicron VOCs

With the emergence of constantly evolving viral variants, researchers are exploring various strategies, such as developing cocktails of multiple monoclonal antibody therapies targeting different regions of the virus, engineering monoclonal antibodies to enhance their potency and breadth of neutralization, and identifying new monoclonal antibody development targets beyond the S protein. These strategies aim to respond to the challenges posed by the mutated virus to currently existing prevention and control measures and to provide a more comprehensive and sustainable solution to control the epidemic effectively. The following text aims to provide a detailed summary and presentation of the neutralizing mAb development strategies being adopted to combat viral evasion of Omicron VOCs.

#### 2.2.1. Optimization of Epitope Selection for SARS-CoV-2

Developing neutralizing mAbs that can effectively neutralize SARS-CoV-2 variants is a major challenge for researchers, given that the future evolution of the virus cannot be predicted. Therefore, it is crucial to develop potent and long-lasting neutralizing mAbs that can target current and emerging SARS-Cov-2 variants, as well as other sarbecoviruses that may appear in the future. 

##### Optimizing the Selection of S Protein Epitopes

Based on the mechanisms of neutralization and differences in potency of human neutralizing antibodies that bind to RBD, researchers have categorized them into four classes [46,47]. The RBD epitopes of Class 1 and 2, which directly overlap with the ACE2 receptor binding site, exhibit less conservation. These epitopes are commonly induced by vaccination or infection [19,48] and are prone to accumulating mutations, which results in frequent viral evasion [46]. Fan et al. [49] identified mAbs that could cross-reactively neutralize sarbecoviruses, including all Omicron variants, by screening single mouse B cells that secreted IgGs binding ≥2 sarbecovirus RBDs. Those neutralizing mAbs were proved to target the class 3 and class 1/4 epitopes, which are more conserved. 

To date, most mAbs that target the RBM of the S protein have limited breadth, meaning they are easily evaded by mutations despite having high neutralization potency [50]. In contrast, non-RBM targeting neutralizing mAb, such as sotrovimab, is still effective against Omicron variants and other sarbecoviruses [41]. Recent research has found that although sotrovimab’s potency is slightly reduced against the Omicron, it still maintains overall neutralizing activity, whereas all RBM-specific mAbs lose their activity [39]. Thus, researchers should prioritize developing mAbs that have high affinity binding and moderate sarbecovirus breadth to increase their resistance to viral evasion [39]. 

Apart from the RBD/RBM, the S2 subunit of the SARS-CoV-2 S protein represents a potential epitope for the development of neutralizing mAbs. The S2 subunit is responsible for fusing the virus to the host cell membrane, allowing the release of the virus genome into the host cell [51]. S2 is relatively conserved in amino acid sequence, and the antibodies it induces are also capable of neutralizing the virus, particularly with broad-spectrum neutralizing activity against a range of coronaviruses [52,53,54]. Most antibodies induced through natural infection targeting the S1 subunit can neutralize the virus, while antibodies with neutralizing activity targeting the S2 subunit are relatively fewer [55]. Piepenbrink et al. [56] suggested an in vivo cooperative effect between human mAbs specific to the S1 subunit and S2 subunit epitopes, leading to broad neutralization of SARS-CoV-2 variants. This makes the S2 region an attractive target for vaccine and antibody therapies that can offer cross-protection against different strains of coronaviruses. 

##### Monoclonal Antibodies That Do Not Directly Neutralize the Virus but Instead Target Human ACE2 Receptors

The RBD of SARS-CoV-2 exhibits high variability through continuous mutation, while human ACE2 receptors remain stable. Targeting ACE2 with monoclonal antibodies can also neutralize the virus by blocking its entry into host cells. This approach has shown potential in preclinical studies as it can target the receptor used by multiple coronaviruses, including SARS-CoV-2. Clinical trials have shown that ACE2 receptor activation has the potential to alleviate pulmonary injury, vascular damage, and lung fibrosis, which can synergistically relieve symptoms of COVID-19 [57]. In a study by Du et al. [58], an ACE2-blocking mAb was isolated and humanized. The mAb was found to possess robust inhibitory activity against both SARS-CoV and circulating global strains of SARS-CoV-2 without any significant changes in blood pressure, hematology, or toxicology parameters in cynomolgus monkeys. These findings suggest that ACE2-blocking monoclonal antibodies hold promise as a safe and effective therapeutic intervention against COVID-19.

##### Block the Other Routes for Viral Entry in Addition to the RBD-ACE2

In addition to the RBD-ACE2, there are other routes for SARS-CoV-2 infection to host cells. Several essential host factors, such as CD147 (Basigin/EMMPRIN) [59], UFO (AXL) [60], TMEM30A [61], LDLRAD3 [62], CLEC4G [62], KREM [63], ASGR1 [63], and NRP1 [64], have been identified as potential therapeutic targets and may serve as alternative functional receptors of the virus. CD147 binds to the RBD of the S protein, while UFO, TMEM30A, LDLRAD3, and CLEC4G interact with NTD, independently mediating viral entry and infection [65]. KREMEN1 and ASGR1 not only bind to the RBD with high potency but also interact with the NTD, acting as an independent viral entry without ACE2 in cells and even in mice. Blocking the interaction of NRP1 with the S1 CendR motif of the S protein has been shown to alleviate SARS-CoV-2 infection [64]. These ACE2-independent receptors may be potential mAbs targets against SARS-CoV-2 regardless of variants. For instance, CD147 is an adhesion molecule that plays a significant role in mediating infection and cytokine storms [66]. It is a signal initiator for cytokine storm, which is closely related to severe COVID-19 infection [67,68] and a mediator of pulmonary fibrosis [69]. Meplazumab, a mAb targeting CD147, has been demonstrated to efficiently block viral entry of SARS-CoV-2 and its variants while alleviating cytokine storm [59,70,71] and progression of pulmonary fibrosis of COVID-19 [69].

##### Take Full Advantage of Non-Neutralizing Antibodies That Target Conserved Epitopes in the S Protein

In the past, the focus of discovering monoclonal antibody treatments for viral infections has been on identifying epitopes that can trigger the production of neutralizing antibodies against the virus. As mentioned earlier, monoclonal antibodies targeting neutralizing epitopes often encounter the issue of viral escape [72]. Non-neutralizing epitopes that are highly conserved are less susceptible to immune pressure as compared to neutralizing epitopes. This is because the virus’s ability to infect cells is not affected by antibody binding at these sites. Cross-reactive, non-neutralizing antibodies, which have typically been disregarded, can also be incredibly effective in developing potent and wide-ranging antiviral agents. 

Lim et al. [73] demonstrated in their study that non-neutralizing antibodies targeting the RBD could enhance the neutralizing antibodies’ ability to deactivate SARS-CoV-2 when linked to neutralizing binders. By combining these non-neutralizing Fabs with neutralizing antibodies in a bispecific VH/Fab IgG format, they witnessed a significant 25-fold increase in potency compared to using only neutralizing antibodies or a combination of neutralizing and non-neutralizing antibodies separately. These findings suggest that bispecific antibodies that target both neutralizing and non-neutralizing epitopes on the RBD could be a valuable and efficient strategy for developing more potent SARS-CoV-2 antibodies.

On the other hand, Weidenbacher et al. [74] developed a type of inhibitor for SARS-CoV-2, named receptor-blocking conserved non-neutralizing antibodies (ReconnAbs), that are capable of neutralizing all variants of concern, including Omicron. These inhibitors were created by attaching the SARS-CoV-2 receptor ACE2 to non-neutralizing antibodies that target highly conserved areas of the S protein. ReconnAbs consist of two primary components: a binding component and an inhibitory component. The former is a non-neutralizing antibody that binds to a conserved site on the S protein with high affinity. The latter is the ACE2 domain, which can be substituted with other neutralizing components such as RBD-directed monoclonal antibodies, ACE2 domains with increased RBD-binding activity [75], or aptamers [76]. Additionally, future ReconnAb designs could also target the interaction with DPP4 [77], which is a receptor for other coronaviruses, expanding their range. They pointed out that ReconnAbs hold promise as potential broad-spectrum therapeutics that can effectively combat emerging pandemic diseases, including SARS-CoV-2.

#### 2.2.2. Enhance Antibody Potency

Strategies to enhance antibody potency include increasing the antibody valency, optimizing the binding affinity, and developing multi-specific or bispecific antibodies, which can help to overcome the ability of the virus to mutate and adapt to immune pressure. In addition, sophisticated bioinformatics tools can be employed to identify key amino acid residues involved in the interaction between an antibody and its target antigen [78,79]. These residues can then be targeted for mutagenesis to enhance antigen-binding affinity and other antibody functionalities [80].

##### Cocktail and Bispecific Antibody

A possible strategy to enhance the neutralizing breadth and combat viral evasion is the use of combination therapy that involves utilizing two or more antibodies targeting diverse epitopes [81]. This approach may minimize mutational escape by SARS-CoV-2, as the simultaneous mutation of two antibodies that bind to distinct and non-overlapping regions of the S protein in a cocktail therapy improves neutralization breadth and creates a higher barrier for viral evasion [34,81].

Generating bispecific antibodies presents another alternative that may offer several advantages, including higher efficacy and reduced costs [82]. Li et al. [83] developed a bispecific antibody, bn03, that synergistically targets two distinct epitopes on the RBD of the Omicron variant using two single-domain antibodies. Thanks to its compact molecular size, bn03 can penetrate deep into the S protein’s trimeric interface when inhaled, making it an effective targeted therapy for highly conserved cryptic epitopes [83]. 

Most of the current multi-specific antibodies target multiple epitopes within the RBD. To expand the target coverage, monoclonal antibodies can be developed and designed to target a combination of the RBD, NTD, and S2 subunits, leading to the generation of multi-specific antibodies that can block viral binding and fusion simultaneously [84].

##### Nanobodies

Camel-derived nanobodies (variable domain of heavy chain of heavy-chain antibody, VHH) have been suggested as a promising approach for reducing the risk of immune evasion caused by the Omicron infection [85]. Nanobodies have remarkable characteristics, including small size, good stability and solubility, and the ability to bind to cryptic epitopes or cavities [86]. These features make nanobodies an ideal candidate for developing universal binders. 

Researchers investigated the potential of isolating promising VHH binders with desired properties from a newly developed synthetic VHH library [87,88]. The synthetic origin of the library provides an advantage in terms of the diversity of epitopes that can be targeted, as compared to human antibodies. By employing a depletion panning strategy to remove binders targeting regions outside the RBM, we were able to generate neutralizing antibodies. Furthermore, the RBD variant was altered in each round of selection, which enabled the generated neutralizing potential to be extended to a broader spectrum of SARS-CoV-2 variants of concern. These findings highlight the potential of utilizing synthetic VHH libraries as a valuable resource for developing effective antibodies against viral pathogens, ultimately helping to minimize the risk of emerging SARS-CoV-2 variants.

Another noteworthy characteristic of nanobodies binding domains is their capacity to form multivalent structures, resulting in a non-linear increase in binding properties [89]. Multivalent nanobodies can resist the mutations of the SARS-CoV-2 variants by two mechanisms: increased avidity for the ACE2 binding domain and identifying preserved epitopes that are not accessible to human antibodies, such as the viral domain covered by glycans [90]. Moreover, bi-specific VHH-Fc antibodies have shown greater efficacy in inhibiting S1 RBD and S/ACE2 of SARS-CoV-2 than monoclonal VHH-Fc antibodies [91]. These findings suggest that nanobodies could be a useful technique for neutralizing SARS-CoV-2 variants, even in the presence of new mutations [90].

##### IgM, IgA, and Polymeric IgG

One potential effective solution to combat viral evasion and improve the effectiveness of antibody-based treatments against viral infections is to enhance neutralizing antibody valency [92] by converting antibody type.

Research has demonstrated that IgM can increase valency and enhance potency by triggering avidity effects [93]. For instance, the naturally decavalent IgM pentamer with repetitive antigen-binding variable fragments [94] can offer increased potency through epitope-dependent steric hindrance as well as engaging more epitopes on the S proteins than an IgG1, which may be related to antibody binding angles [46]. The IgG versions of antibodies seem to have lower avidity than their IgM counterparts, and meanwhile, the neutralizing potency of the antibodies may decrease after an artificial switch from IgM to IgG [95]. While converting IgG into IgM does improve antibody potency, selecting the most effective IgM capable of overcoming resistance relies heavily on identifying and targeting the optimal epitope.

Dimeric IgA in mucosal tissues is approximately 15-fold more effective against SARS-CoV-2 than monomeric IgG [96]. Recent studies have shown that individuals with elevated levels of the Wildtype S protein-specific IgA in their nasal swabs exhibit reduced Omicron breakthrough infections and a lower viral load [97]. It is possible that secretory IgA could provide broader cross-reactivity against variants [98]. In a study by Wang et al. [99], the administration of an intranasal Ad5-S-Omicron booster, in combination with ancestral vaccines, has been shown to establish robust mucosal and systemic immunity against Omicron variants.

Furthermore, Zou et al. [100] constructed a polymeric P5-22 IgG with 12 Fabs, which exhibits resistance to viral mutational escape. The polymeric P5-22 is capable of restoring P5-22 binding to all F486-mutated RBDs and neutralizing F486V and F486R effectively. They also suggested that the parental IgG should have at least a weak binding affinity for mutated epitopes for the polymeric IgG to overcome viral mutational escape.

The above studies suggest that multivalent antibodies have advantages in resisting viral escape. However, the technical barriers for isolating or modifying specific antibody types should not be overlooked, as well as the potentially higher production costs.

#### 2.2.3. Optimizing Immunization Regimens, such as Antigen Components, Immunization Routes, and Adjuvants

One promising approach worth exploring is to induce stronger, more persistent, and cross-protective immune responses against Omicron subvariants by refining immunization regimens, such as optimizing antigen composition, immunization routes, and adjuvants, while, subsequently, acquiring neutralizing mAbs with corresponding properties.

Fan et al. [49] successfully generated neutralizing mAbs that were cross-reactive against conserved sarbecovirus RBD epitopes through immunization with nanoparticles co-displaying RBDs of the S protein from eight sarbecoviruses. This immunization approach, which involves the co-display of multiple neutralizing epitopes of SARS-CoV-2 variants and other sarbecoviruses, is one of the strategies for developing broadly effective neutralizing antibodies.

It is well-known that the upper respiratory tract is the most common site for SARS-CoV-2 infection [101]. Intranasal vaccines offer the valuable benefit of inducing robust tissue-resident memory B and T cell responses within the respiratory system, thereby providing swift and potent primary defense against mucosal pathogens [102,103]. Cai et al. [104] conducted a study that demonstrated the effectiveness of one or two intranasal boosts of the fragment crystallizable (Fc)-linked RBD-binding domain derived from the wild-type SARS-CoV-2 (Wuhan-Hu-1) in inducing significantly higher neutralizing antibodies against Omicron subvariants, including BA.5.2 and XBB.1. The combination of intramuscular priming and intranasal boosting can provide broader cross-protection against Omicron variants and subvariants, with the potential to prolong the interval needed for updating the vaccine immunogen from months to years [104,105]. On the other hand, the addition of the Fc fragment helps the vaccine bind to and be taken up by antigen-presenting cells (APCs). This can potentially prevent the vaccine from being broken down by proteases in the body, thus, increasing its duration of effectiveness [106]. In addition, the interaction between the Fc fragment and APCs also boosts the vaccine’s ability to stimulate the immune system’s response to the RBD, resulting in a more sustained immune activation [106,107].

Moreover, adjuvants can also improve the immune response to antigens during the immunization process [108], thereby increasing the likelihood of producing more specific and effective neutralizing mAbs with high affinity against the target antigen. Liu et al. [109] have developed a pan-sarbecovirus vaccine composed of the RBD from the original SARS-CoV-2 strain along with a novel adjuvant CF501, an agonist of stimulator of interferon genes (STING), which help induce broad-spectrum neutralization against diverse variants of SARS-CoV-2, including BQ.1.1 and XBB. This research sheds light on neutralizing mAb development, showing that appropriate adjuvant could enhance the induction of broad-spectrum neutralizing antibodies for SARS-CoV-2 vaccines.

## 3. Conclusions

The existing mAbs-based therapy has shown a significant decrease in their neutralizing activity against the current Omicron variants, and in some cases, their effectiveness has been completely lost. In response to this challenge, future strategies for the development of mAbs must focus on improving their resilience against viral evasion. One strategy involves optimizing the targeted epitope, which includes epitopes within and outside the S protein, or even blocking viral entry directly by binding to the host receptor. Another strategy is to enhance both the affinity and potency of mAbs. Moreover, non-neutralizing antibodies that bind to conserved epitopes on the S protein should be fully utilized. Finally, the immunization regimen should be optimized to maximize the effectiveness of mAbs.

## Figures and Tables

**Figure 1 viruses-15-01297-f001:**
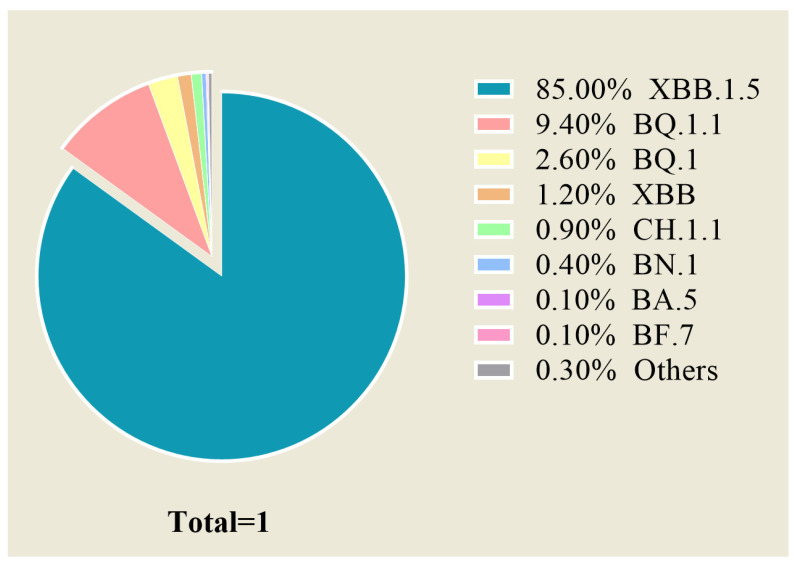
Omicron circulating variants in the USA. The above data are current as of 23 February 2023 [31].

## Data Availability

Not applicable.

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
