# Peer review of "Status and Developing Strategies for Neutralizing Monoclonal Antibody Therapy in the Omicron Era of COVID-19"

_viruses, 2023, doi:10.3390/v15061297_

Round 1

Reviewer 1 Report

In this review, the authors present and highlight the amino acid sequence of RBD of S protein and the key mutations of therapeutic epitopes/ binding sites of the S protein for different variants of SARS-CoV-2, especially, the Omicron. Then they highlight the challenge of the monoclonal antibodies of SARS-CoV-2, followed by summarizing the strategies that were developed to improve the neutralization of mAbs, such as increasing the binding affinity, optimization of the targeting epitopes of SARS-CoV-2, incorporation of using non-neutralizing antibodies to bind with conserved epitopes of the S protein, optimization of immunization regimens.  

Some suggestions: Provide a paragraph to summarize and discuss the advantages and disadvantages of the different categories of monoclonal antibodies.

Line 211: SARS-Cov-2 > SARS-CoV-2

Line 211: Correct: ACE2-indepandent

Line 93: reference 32 is not correct. Double-check the order of references.

minor revision

Author Response

We have carefully considered the suggestion of Reviewer and make some changes.

Responds to the reviewers' comments:

1、Some suggestions: Provide a paragraph to summarize and discuss the advantages and disadvantages of the different categories of monoclonal antibodies.

Thank you very much for your advice. We have added the following paragraph to the end of the section "IgM, IgA and polymeric IgG":  The above studies suggest that multi-valent antibodies have advantages in resisting viral escape. However, the technical barriers for isolating or modifying specific antibody types should not be overlooked, as well as the potentially higher production costs.

2、Line 211: SARS-Cov-2 > SARS-CoV-2

We have corrected it according to your suggestion.

3、Line 211: Correct: ACE2-indepandent

We have replaced it with "ACE2-independent"

4、Line 93: reference 32 is not correct. Double-check the order of references.

Sorry, there may have been an error when inserting references as I was preparing another article on Orthopoxviruses at the same time of writing this review. We have corrected it.

Reviewer 2 Report

The manuscript 'Status and Developing Strategies for Neutralizing Monoclonal Antibody Therapy in the Omicron Era of COVID-19' drafted by Jie Peng and colleagues have detailed the benefits, shortages, barriers and future directions etc. of neutralizing monoclonal antibodies. The whole review has briefly summarized most information about the NmAb since the first outbreak of pandemic in early 2020. Even SARS-CoV-2 is no longer characterized as a global health crisis, and daily life is returing to normal. But the new variants are still emerging, and the harmness and casualties caused by this virus can still not be ingored. Thus, develping more broadly and effective NmAb is definitely worthy to invest labors and funding in nowadays and in the future. This review will help the researchers hold the right directions for future investigation. 

I do think this manuscript will be suitable for publication after minor revision. Below are the indicated minor issues needed to be corrected.

1. There should be a space beteween the text and the reference, which should be consistent in the whole manuscript. Other formatting errors should also be corrected thoroughly. 

2. Some misspelling or mistakes existed, like: line '147', 'amis' should be changed with 'aims'; line '210', 'in even mice' should be corrected with 'even in mice'.

3. From line '366' to line '369', based on the quoted paper (reference 108), only the adjuvant can not induce broad-spectrum neutralization against diverse variants of SARS-CoV-2 or even the neutralizing mAbs. The adjuvant can only help improve such effects.

Author Response

We have carefully considered the suggestion of Reviewer and make some changes.

Responds to the reviewers' comments:

1.There should be a space beteween the text and the reference, which should be consistent in the whole manuscript. Other formatting errors should also be corrected thoroughly.

We have corrected the formatting errors as per your helpful reminder.

2. Some misspelling or mistakes existed, like: line '147', 'amis' should be changed with 'aims'; line '210', 'in even mice' should be corrected with 'even in mice'.

Thank you for your reminder, and we have made corrections to the paragraph. 

3. From line '366' to line '369', based on the quoted paper (reference 108), only the adjuvant can not induce broad-spectrum neutralization against diverse variants of SARS-CoV-2 or even the neutralizing mAbs. The adjuvant can only help improve such effects

Thank you for pointing out the mistake. We have modified the statement to: Zezhong Liu et al.[110] have developed a pan-sarbecovirus vaccine composed of the RBD from the original SARS-CoV-2 strain along with a novel adjuvant CF501, an agonist of stimulator of interferon genes (STING), which help induce broad-spectrum neutralization against diverse variants of SARS-CoV-2, including BQ.1.1 and XBB. This research sheds light on neutralizing mAb development, showing that appropriate adjuvant could enhance the induction of broad-spectrum neutralizing antibodies for SARS-CoV-2 vaccines..